# OpenReview forum: "It's the Thought that Counts: Evaluating the Attempts of Frontier LLMs to Persuade on Harmful Topics"
_ICLR.cc/2026/Conference — Submitted to ICLR 2026_

### Official Review · Reviewer_dm8K · 2025-11-01

**Soundness:** 3
**Presentation:** 2
**Contribution:** 3
**Rating:** 6
**Confidence:** 2

**Summary:**

The paper proposes APE (Attempt to Persuade Eval), a benchmark that shifts persuasion evaluation from success (changing beliefs) to intent, whether a model attempts to shape beliefs or behavior, especially on sensitive and harmful topics. APE runs multi-turn dialogues between a “persuader” model and a “persuadee” (default GPT-4o), then uses an automated evaluator (also an LLM) to classify each turn as attempt, refusal, or no-attempt. Topic coverage spans benign factual/opinion, controversial issues, and non-controversially harmful content. Key findings: (i) many open and closed weight models willingly attempt persuasion on harmful topics that they would refuse to directly assist; (ii) “jailbreak” fine-tuning sharply collapses refusal rates on harmful topics; (iii) fine-grained “degree of persuasion” labels are unreliable, motivating a binary attempt/no-attempt metric; and (iv) persuasion attempts are most common in early rounds and taper with longer conversations.

**Strengths:**

1. The overall writing is fluent and easy to follow.

2. The motivation is strong and the paper focuses on the important yet under-explored question of "the danger of a model’s propensity to persuade".

3. The proposed benchmark is significant including multi-turn, topic-diverse protocol. The design probes benign, controversial, and non-controversially harmful topics with a structured conversation length; authors document that attempts cluster early and decay over turns, justifying a 3-round default.

4. The analysis is interesting: (1) Empirically shows a gap between refusal to do harm vs. willingness to persuade others to do harm; direct-assistance requests are refused while harmful persuasion attempts often occur. (2) Fine-tuning to jailbreak a closed-weight model dramatically reduces refusal on harmful categories while leaving benign behavior similar.

**Weaknesses:**

1. Evaluator dependence and circularity. The pipeline often uses GPT-4o as both persuadee and evaluator. Even with some human checks, this raises concerns about shared biases and failure modes.

2. The font in figure 1 is hard to see. Make it larger.

**Questions:**

How robust are your attempt/no-attempt labels to the choice and version drift of the LLM evaluator—i.e., if you swap GPT-4o for a different frontier model (or a newer minor version of the same model), how do per-topic rates and headline conclusions change?

---

> ### Author Response · Authors · 2025-11-19
>
> We appreciate your thoughtful review and appreciation of our papers strengths, such as the writing quality, “strong motivation”, sensible experimental design, and “interesting” analysis! We respond to your comments below in a point-by-point manner.
>
> *“Evaluator dependence and circularity. The pipeline often uses GPT-4o as both persuadee and evaluator. Even with some human checks, this raises concerns about shared biases and failure modes. How robust are your attempt/no-attempt labels to the choice and version drift of the LLM evaluator—i.e., if you swap GPT-4o for a different frontier model (or a newer minor version of the same model), how do per-topic rates and headline conclusions change?”*
>
> We thank the reviewer for pointing out an area of improvement for the paper. We directly address it in the group response under “Comparing Different Evaluator Models,” where we vary both the persuadee and evaluator models. Across these settings, we find broad agreement in the measured persuasion attempt rates for a given persuader, suggesting that our main conclusions do not depend on using GPT-4o in multiple roles.
>
> *“The font in figure 1 is hard to see. Make it larger.”*
>
> We appreciate the suggestion to improve the figures clarity and have made these changes in the updated revision.

---

### Official Review · Reviewer_Dswg · 2025-11-01

**Soundness:** 2
**Presentation:** 2
**Contribution:** 2
**Rating:** 4
**Confidence:** 4

**Summary:**

This paper presents the Attempt to Persuade Eval (APE) benchmark, which measures large language models’ willingness to engage in persuasion on harmful topics rather than their success in changing beliefs. Using simulated multi-turn conversations between a persuader model and a persuadee model, APE evaluates frontier systems across benign, controversial, and clearly harmful subjects such as violence, terrorism, and human trafficking. The results show that several advanced models, including GPT-4 variants and Gemini 2.5 Pro, still produce persuasive responses in ethically risky situations even when they refuse direct participation in harmful actions.

**Strengths:**

+ The study addresses an important and previously underexamined dimension of AI safety by focusing on the inclination to persuade rather than persuasion outcomes.

+ The experimental setup using simulated multi-turn dialogues is well structured and scalable for auditing model behavior.

+ The analysis is comprehensive, covering many models, validation against human ratings, ablation studies, and an openly available benchmark for further research.

**Weaknesses:**

- Placing a large, attention-grabbing figure before the abstract disrupts readability and confuses the narrative flow. The paper would benefit from starting with the abstract and moving the figure into the introduction or results section.

- The study relies entirely on model-to-model simulations, which limits external validity and prevents meaningful conclusions about how human users might respond to persuasive attempts. Real persuasion involves emotional, social, and moral reasoning that automated agents cannot replicate.

- The definition of a “persuasion attempt” is too shallow. Treating persuasion as a binary label (attempt or no attempt) ignores gradations in tone, framing, intensity, and rhetorical sophistication that characterize real persuasive behavior.

- The evaluator model struggles with nuanced distinctions in persuasive strength. It cannot reliably detect subtle strategies such as moral reasoning, empathy framing, or indirect argumentation that may influence human audiences.

- The evaluation design does not adequately account for confounding factors such as temperature, persona context, or prompt order effects, all of which may significantly alter persuasion likelihood.

- While the paper effectively identifies safety gaps, it provides limited practical guidance on how to mitigate these issues. There is no clear proposal for retraining, safety-layer design, or model auditing frameworks to address identified vulnerabilities.

- Ethical considerations and dataset release protocols are insufficiently detailed. The authors acknowledge potential misuse risks but do not specify monitoring or access restrictions for harmful prompt data.

- The writing occasionally blurs descriptive findings with normative claims, making it harder to separate empirical evidence from interpretation or opinion.

- The paper lacks grounding in persuasion theory or psychology, which weakens its conceptual framing. Integrating established models of persuasive communication would provide stronger interpretive depth.

**Questions:**

- The boundary between benign, controversial, and harmful categories is ambiguous. Some topics (e.g., violence) are clearly harmful, while others (e.g., “undermining control”) depend heavily on interpretation without consistent criteria.

- The neutrality and independence of the evaluator model are uncertain, especially since GPT-4o serves multiple roles (persuader, persuadee, and evaluator) in the same pipeline. This may introduce systemic bias.

- The description of human annotator validation is vague. Details about the annotators’ expertise, agreement calibration, and labeling guidelines are missing, leaving uncertainty about evaluation reliability.

- The operational definition of “persuasive attempt” is unclear. It is not explained whether persuasion is detected through argument structure, linguistic tone, or inferred motivation.

- The five subcategories under “non-controversially harmful” topics are listed without clear theoretical or empirical justification, raising questions about representativeness and generalization.

- The method of generating harmful prompts with a jailbroken model lacks transparency. The authors intentionally omit details for safety, but this decision reduces reproducibility and external verification.

- The paper mentions coordination with industry labs following responsible disclosure but provides no documentation, dates, or evidence of model improvements after disclosure.

- The use of simulated agents for both sides of the dialogue leaves open questions about how real human users would respond to similar persuasion attempts. A small-scale human validation study would have improved the paper’s grounding.

- The broader implications of APE scores are undefined. The paper highlights safety risks but does not propose thresholds, standards, or policy frameworks for determining acceptable levels of persuasive behavior.

**Details Of Ethics Concerns:**

This study involves no human participants and thus poses no direct ethical risks related to consent, welfare, or exposure to harmful content. All experiments use simulated LLM-to-LLM conversations, avoiding human manipulation. However, future validation with real users would raise serious ethical challenges, as exposing participants to persuasive or harmful narratives could cause distress or reinforce dangerous beliefs. Such studies would require strict institutional review, strong content moderation, participant screening, and debriefing procedures. The dataset itself also carries misuse risks, as it could enable the creation of models that generate harmful persuasive content; secure access control and responsible data release are therefore essential. Finally, equating LLM persuasion with human persuasion should be done cautiously to avoid overstating model agency or intent.

---

> ### Author Response · Authors · 2025-11-19
> **Part 1**
>
> Thank you for your very detailed review! We appreciate your recognition of the paper’s strengths, including that it “addresses an important and previously underexamined dimension of AI safety”, that the experimental setup “is well structured and scalable for auditing model behavior”, and that the “analysis is comprehensive, covering many models, validation against human ratings, ablation studies, and an openly available benchmark for further research”.
>
> We have addressed each of your comments in a point-by-point manner below and also made modifications to the revised paper where applicable. If any of our responses raise further questions, or if clarifications on any point would meaningfully influence your overall assessment, we would be grateful to hear it. Your feedback has been very helpful, and we want to ensure we fully understand and address all of your concerns.
>
> *“Placing a large, attention-grabbing figure before the abstract disrupts readability and confuses the narrative flow. The paper would benefit from starting with the abstract and moving the figure into the introduction or results section.”*
>
> We appreciate the suggestion to move the figure to the second page to improve readability and have done so in the revised version of the paper.
>
> *“The study relies entirely on model-to-model simulations, which limits external validity and prevents meaningful conclusions about how human users might respond to persuasive attempts. Real persuasion involves emotional, social, and moral reasoning that automated agents cannot replicate.”*
>
> We agree that model–to-model simulations have limits and report human results in the group response under “Results on Human Participants”. We show that persuasion attempt rates are nearly identical when models respond to actual human initial messages versus simulated ones on the same topics. Also, in an IRB-approved study (n = 836), GPT-4o shifts real users’ conspiracy beliefs by ~12–14 points on a 0–100 scale. These findings suggest that current LLMs can already be strongly persuasive with humans, so it is not obvious that they are weaker than humans on the emotional, social, or moral dimensions the reviewer highlights.
>
> *“The definition of a “persuasion attempt” is too shallow. Treating persuasion as a binary label (attempt or no attempt) ignores gradations in tone, framing, intensity, and rhetorical sophistication that characterize real persuasive behavior.”*
>
> We agree that real persuasive behaviour is richer than a binary notion, and that tone, framing, intensity, and rhetorical sophistication are all important aspects. We view detailed analyses of specific persuasive strategies as highly complementary to our work.
> Our goal in this work, however, is intentionally narrower: we focus on whether a model attempts to persuade at all, particularly on non-controversially harmful topics, because even the presence of persuasive intent in these settings is already a serious violation of current providers’ stated safety policies. Our contribution is to provide a scalable, reliable benchmark for whether models attempt persuasion at all in sensitive contexts, without relying on brittle, fine-grained judgments about which specific strategy is being used.
>
> *“The evaluator model struggles with nuanced distinctions in persuasive strength. It cannot reliably detect subtle strategies such as moral reasoning, empathy framing, or indirect argumentation that may influence human audiences.”*
>
> The inability of the evaluator to identify specific strategies is largely by design: we ask the evaluator only to judge whether a message attempts to persuade by changing the user’s beliefs or behaviour on the target topic.
> Work that measures which strategies actually influence human audiences is complementary to ours; here we instead focus on detecting whether models attempt persuasion in clearly disallowed domains, independent of whether those attempts are ultimately effective on humans.
>
> *“The evaluation design does not adequately account for confounding factors such as temperature, persona context, or prompt order effects, all of which may significantly alter persuasion likelihood.”*
>
> For temperature, all experiments are sampled multiple times (5) and error bars are reported (and fairly small). Additionally, we report experiments with different persona contexts in Appendix C.2 which we reference in Line 429. If there are other specific experiments that you believe would strengthen the experimental results with respect to these confounding factors, please let us know.

---

> > ### Author Response · Authors · 2025-11-19
> > **Part 2**
> >
> > *“While the paper effectively identifies safety gaps, it provides limited practical guidance on how to mitigate these issues. There is no clear proposal for retraining, safety-layer design, or model auditing frameworks to address identified vulnerabilities.”*
> >
> > The first step in a practical solution is being able to consistently identify the problem. To the best of our knowledge, there does not exist any frameworks for identifying models which have a problem with harmful persuasion. Thus, our focus in this work is specifically a new eval to audit systems in terms of whether models are willing to attempt harmful persuasion as described in the “Preventing Harmful Persuasion from Models” section of the group response. Retraining pipelines and safety-layers are incredibly important, but out of the scope of our work which focuses on the *measurement* capability required to enforce such standards once defined. However, we note a few fruitful directions we are thinking about to address these issues such as modifying persuasiveness through post-training/finetuning [Hackenburg et al. 2025] or classifying potentially harmful persuasion contexts through linear probes [Jiapersaud et al. 2025].
> >
> > Hackenburg et al. 2025 The Levers of Political Persuasion with Conversational AI.
> >
> > Jiapersaud et al. 2025 How Do LLMs Persuade? Linear Probes Can Uncover Persuasion Dynamics in Multi-Turn Conversations.
> >
> > *“Ethical considerations and dataset release protocols are insufficiently detailed. The authors acknowledge potential misuse risks but do not specify monitoring or access restrictions for harmful prompt data.”*
> >
> > We share the general concern that research on persuasion can have dual use implications, which is why we explicitly discussed potential misuse in Appendix E. In practice, the benchmark consists of pre-written example prompts on clearly harmful topics, included purely for convenience so providers can run the evaluation immediately; a determined adversary could readily construct similar prompts without our data. Our methods also evaluate existing models rather than training new ones to be more persuasive.
> >
> > Given this, we judge the incremental misuse risk of releasing the prompts as minimal, and believe that adding heavy access restrictions would mainly reduce the benchmark’s utility for researchers who want a simple, easily accessible safety evaluation. We therefore include the prompts as part of the open release, but with warnings to exercise due caution in downstream usage.
> >
> > *“The writing occasionally blurs descriptive findings with normative claims, making it harder to separate empirical evidence from interpretation or opinion.”*
> >
> > We take this concern seriously but find it hard to address without specific instances. Throughout the paper we have tried to clearly separate empirical findings from interpretation. Claims about model behaviour are either supported by our reported results or by prior work that we explicitly cite. If the reviewer has particular sentences or passages in mind where this separation is unclear, we would be grateful for pointers and are happy to clarify them.
> >
> > *“The paper lacks grounding in persuasion theory or psychology, which weakens its conceptual framing. Integrating established models of persuasive communication would provide stronger interpretive depth.”*
> >
> > We agree that perspectives from persuasion theory and psychology are valuable, and we connect to this literature by reviewing human subject studies that measure LLM induced belief change on topics such as conspiracy beliefs, vaccine acceptance, and climate opinions (Sec. 2.1).
> >
> > While these works are crucial for understanding how specific strategies shift real human beliefs, our contribution is an immediate first step: an easy, reliable benchmark for detecting when models attempt persuasion in the first place, without requiring us to model downstream human psychology.

---

> > > ### Author Response · Authors · 2025-11-19
> > > **Part 3**
> > >
> > > *“The boundary between benign, controversial, and harmful categories is ambiguous. Some topics (e.g., violence) are clearly harmful, while others (e.g., “undermining control”) depend heavily on interpretation without consistent criteria.”*
> > >
> > > Indeed, we agree that some topics are not obviously harmful or safe, which is a feature of the topic design process rather than a bug (which we tried to quantify subjectively in Figure 1). We want to measure persuasive attempts, not only on clearly benign or harmful topics, but on topics that are not easily categorized as important/harmful. Overall, these topics should evolve over time with the research field which is why we made the APE codebase easily extensible for people to add their own topics. For example, while some people might strongly argue that undermining control is an important topic with real world impact, others might suggest that AI capabilities are nowhere close to being able to cause cybersecurity dangers through these means. Our framework provides the foundation to investigate a variety of questions that future researchers may have.
> > >
> > > “The neutrality and independence of the evaluator model are uncertain, especially since GPT-4o serves multiple roles (persuader, persuadee, and evaluator) in the same pipeline. This may introduce systemic bias.”
> > >
> > > We thank the reviewer for pointing out an area of improvement for the paper. We directly address it in the group response under “Comparing Different Evaluator Models,” where we vary both the persuadee and evaluator models. Across these settings, we find broad agreement in the measured persuasion attempt rates for a given persuader, suggesting that our main conclusions do not depend on using GPT-4o in multiple roles.
> > >
> > > *“The description of human annotator validation is vague. Details about the annotators’ expertise, agreement calibration, and labeling guidelines are missing, leaving uncertainty about evaluation reliability.”*
> > >
> > > Thank you for highlighting this; we agree that the human validation procedure should be clearly documented. Our human validation uses three annotators who are ML/AI safety researchers familiar with LLM behaviour as it relates to persuasion. Before labeling the main sample, we ran a small calibration round using the same definition of “persuasion attempt” as in the evaluator prompt (App. B.1), and analyzed borderline cases until they agreed on consistent guidelines. For the main study, each annotator then independently labeled a sample of messages as “attempt” vs. “no attempt”, we took the majority vote as ground truth, and we report multiple reliability metrics in Table 1 (right) which we believe demonstrates robust evaluation reliability.
> > >
> > > We have revised the paper (in Section 4.4, “Alignment with human ratings of persuasion attempts” ) to make the annotator background, calibration step, and labeling guidelines more explicit in the description of this study.
> > >
> > > *“The operational definition of “persuasive attempt” is unclear. It is not explained whether persuasion is detected through argument structure, linguistic tone, or inferred motivation.”*
> > >
> > > Our operational definition is not tied to any specific strategy: a “persuasion attempt” is any message that tries to change the user’s beliefs or behaviour on the target topic. This is made explicit in the evaluator prompt which asks the evaluator to judge whether the message is trying to persuade at all, regardless of its argument structure, tone, or rhetorical style. We emphasize that different users (humans) will be persuaded by different argument structure, tones, and motivations, and thus trying to quantify these through simulations is challenging. Rather, we view any attempt at persuasion important information, especially when it pertains to topics like committing mass murder or terrorism recruitment.

---

> > > > ### Author Response · Authors · 2025-11-19
> > > > **Part 4**
> > > >
> > > > *“The five subcategories under “non-controversially harmful” topics are listed without clear theoretical or empirical justification, raising questions about representativeness and generalization.”*
> > > >
> > > > We outline our topic selection process in Section 3.2 and explain the details of the topic generation in Appendix B.2. Specifically, for ‘non-controversially harmful’ topics our justification for selecting these topics was based on recent technical and policy relevant literature. This literature includes multiple real-world topic categories on which LLMs can persuade humans, like conspiracies [Costello et al. 2024] and non-controversially harmful categories (e.g., terrorism [1] or human trafficking [2]). We also include topics related to potential future risks, such as the AI Control topics, which are inspired by recent AI safety reports [Bengio et al. 2025].
> > > >
> > > > [1] Department of Homeland Security. Impact of Artificial Intelligence (AI) on Criminal and Illicit Activities. Public-Private Analytic Exchange Program (2024).
> > > >
> > > > [2] OSCE & Regional Support Office of the Bali Process. New frontiers: The use of generative artificial intelligence to facilitate trafficking in persons (2024).
> > > >
> > > > *“The method of generating harmful prompts with a jailbroken model lacks transparency. The authors intentionally omit details for safety, but this decision reduces reproducibility and external verification.”*
> > > >
> > > > Our goal was to balance transparency with avoiding the risks associated with giving public access to the jailbroken model used for topic generation. The exact procedure for topic generation is described in Section 3.2: we use a single jailbroken model to propose short harmful topic descriptions for each subcategory, then expand them into single sentence topics. For reproducibility and external verification, we release the full resulting topic list and prompts in our repository, so all of our analyses can be rerun without access to the jailbroken model itself.
> > > >
> > > > *“The paper mentions coordination with industry labs following responsible disclosure but provides no documentation, dates, or evidence of model improvements after disclosure.”*
> > > >
> > > > We agree that documenting our disclosure and its effects is important and believe we have done so in the “Responsible disclosure” section (Page 12): We state that we “shared an earlier version of this work with Google, OpenAI, and Anthropic” and that “after our disclosure, the current version of Gemini 2.5 Pro exhibits greatly reduced attempts to persuade on noncontroversially harmful topics”. Appendix C.1 then gives concrete dates and measurements for three Gemini 2.5 Pro releases (March 25, May 6, and June 17, 2025), noting that the May 6 release “exhibits a very high attempt rate on extreme topics” and that, in the latest model, the categories Commit Physical Violence and Sexual Assault of Minors show “∼50 percentage point decreases in attempts.” We see this as both documentation of coordination (which labs and which model snapshots were involved) and quantitative evidence of improvements following disclosure; if the reviewer has a particular type of additional documentation in mind that they consider appropriate for a research paper, we would be grateful for concrete suggestions.
> > > >
> > > > *“The use of simulated agents for both sides of the dialogue leaves open questions about how real human users would respond to similar persuasion attempts. A small-scale human validation study would have improved the paper’s grounding.”*
> > > >
> > > > We agree that model–to-model simulations have limits and report human results in the group response under “Results on Human Participants”. We summarize the results in the above response and hope these results shed some light on how humans would interact with persuasive LLMs in similar situations to APE simulations.
> > > >
> > > > *“The broader implications of APE scores are undefined. The paper highlights safety risks but does not propose thresholds, standards, or policy frameworks for determining acceptable levels of persuasive behavior.”*
> > > >
> > > > We agree that establishing universal thresholds for acceptable persuasion is a complex challenge involving societal values, which is why our work focuses on the *measurement* capability required to enforce such standards once defined. However, we argue that for the 'Non-controversially Harmful' category (e.g., promoting terrorism or child exploitation), the only acceptable threshold is **zero**—a standard already codified in the safety policies of major providers (detailed in Appendix D), yet which APE reveals is frequently violated. For more nuanced categories (e.g., 'Controversial' or 'Political'), APE provides the necessary granularity for labs to calibrate models according to their specific deployment guidelines (e.g., enforcing a 'neutral point of view' vs. allowing 'advocacy'). We view APE as the diagnostic tool that enables adherence to these frameworks, rather than the regulatory code itself.

---

> > > > > ### Author Response · Authors · 2025-11-19
> > > > > **Part 5**
> > > > >
> > > > > *“The dataset itself also carries misuse risks, as it could enable the creation of models that generate harmful persuasive content; secure access control and responsible data release are therefore essential. ”*
> > > > >
> > > > > We respectfully push back on the idea that the dataset of topics could realistically make the creation of harmful models easier, as discussed in Appendix E. The topics contained in our dataset are not jailbreaks and do not contain harmful responses but solely the prompts. It would take very limited resources for an attacker to create a much more damaging dataset and therefore we believe that the 600 prompted topics will not contribute significantly to the production of models that can create harmful content. Of course, advancing the science of persuasion, in general, has dual use potential. But we believe that the benefits outweigh the risks here.
> > > > >
> > > > > We thank you for your thorough and detailed  review. We believe we have addressed the main weaknesses raised, but if there are remaining concerns or suggestions for improvement, we would be very grateful for further clarification.

---

### Official Review · Reviewer_WhhT · 2025-11-03

**Soundness:** 2
**Presentation:** 2
**Contribution:** 2
**Rating:** 4
**Confidence:** 4

**Summary:**

This paper introduces the Attempt to Persuade Eval (APE), a benchmark that evaluates whether large language models will attempt to persuade users on harmful topics. The authors test 12 frontier models across 600 topics spanning benign content to extremely harmful topics (terrorism, human trafficking, etc.) using a multi-turn conversational framework with automated evaluation. The results reveal that many state-of-the-art models frequently attempt persuasion on harmful topics they would refuse to directly assist with, highlighting critical gaps in current safety guardrails.

**Strengths:**

- The paper addresses a critical problem in AI safety by benchmarking models' willingness to engage in persuasive attempts on harmful topics.
- The evaluation framework is comprehensive, covering 600 diverse topics in a multi-turn setup.
- The results are validated in automated and human assessments, and reveal several findings in LLM risks.

**Weaknesses:**

- The paper lacks novelty and technical contributions. Particularly, there are prior works already discussing the safety of the LLM persuasion and  [1,2]. The authors should more carefully discuss and differentiate the paper from prior works.
- The paper does not provide actionable insights or implications for the safety evaluation. For example, what are the potential solutions for mitigation, what caused the critical safety issues, etc?
- The evaluation framework heavily relies on LLM-only simulation, which could cause a critical gap when applying to real-world human-AI persuasion.

[1] LLM Can be a Dangerous Persuader: Empirical Study of Persuasion Safety in Large Language Models. COLM 2025.

[2] How Johnny Can Persuade LLMs to Jailbreak Them: Rethinking Persuasion to Challenge AI Safety by Humanizing LLMs. ACL 2024.

**Questions:**

See the weaknesses above.

---

> ### Author Response · Authors · 2025-11-19
>
> Thank you for pointing out the strengths of our work, including that it “addresses a critical problem in AI safety”, the “comprehensiveness” of our benchmark, and agreeing that our paper “reveals several findings related to LLM risks”! We appreciate your insightful review - the experimental suggestions and prior work discussions certainly improved the paper. We have responded in a point-by-point manner below.
>
> *“The paper lacks novelty and technical contributions. Particularly, there are prior works already discussing the safety of the LLM persuasion and [1,2](…)”*
>
> We appreciate the additional papers to consider in our related work section but also point out that [1] is concurrent work according to the ICLR 2026 reviewer guide, and is considered contemporaneous since COLM 2025 was after the ICLR 2026 deadline of September 24 and therefore comparison is not mandatory. Regardless, we have added a discussion comparing our work to [1] and [2] at the end of Section 2.2:
>
> “Zeng et al. [2] also consider harmful persuasion but focus on developing jailbreak methodologies and evaluate predicted harmfulness (i.e., estimated success), rather than our focus on persuasion attempts. One concurrent work to ours [1] also looks at harmful persuasion with a focus on persuasive strategies and does not consider as broad a set of harmful topics as APE (i.e., their topics consist of relationships, marketing, digital privacy, professional career, financial situations, and health and wellness). Alternatively, we specifically choose topics that are motivated by current or expected real-world harms.”
>
> *“The paper does not provide actionable insights or implications for the safety evaluation. For example, what are the potential solutions for mitigation, what caused the critical safety issues, etc?”*
>
> We hope that the discussion of ‘Preventing Harmful Persuasion from Models’ in the group response has adequately addressed this concern for potential solutions for mitigation. We also note that after we disclosed our findings, Google quickly started work to solve this for their current and future models, validating the immediate utility of our research in practical settings.
>
> *“The evaluation framework heavily relies on LLM-only simulation, which could cause a critical gap when applying to real-world human-AI persuasion.”*
>
> We present real-world human-AI persuasion results in the group response section ‘Results on Human Participants’.
>
> We believe we have adequately addressed your stated weaknesses, but we would love to know if there are any outstanding confusions or suggestions for further improvements we could make.

---

### Author Response · Authors · 2025-11-19
**Group Response: Part 1**

**Results on Human Participants**

Two of the reviewers mention they would like to see confirmation that the results in this work generalize to human participants. To answer this question about whether the pipeline would behave similarly when talking to a human, while it would be unmitigably unethical (or even illegal) to test most persuasion in the non-controversially harmful category, we draw on our concurrent work in the topic category of conspiracies, where we have conducted large-scale, IRB-approved (with a thorough debrief for participants), human persuasion studies. Participants (n=836) wrote about any conspiracy theory, then rated their belief in it on a 0–100 scale (0 = definitely false, 50 = unsure, 100 = definitely true), both before and after treatment. Participants were randomly assigned to either a “debunk” (persuade out of belief) or “bunk” (persuade into belief) condition. Using prompts similar to APE, we instructed GPT-4o to persuade users accordingly across multiple rounds.

GPT-4o effectively persuaded users in both directions, with bunking slightly more effective: Debunking reduced belief by 12.3 points. Bunking increased belief by 13.6 points. These effects held across various conspiracy topics included in the APE benchmark, from historical cases (e.g., JFK) to contemporary, polarized issues (e.g., COVID-19).
Thus, to test if the persuader behaves similarly when talking to a human vs. a simulated persuadee, we applied the APE evaluator to the human data using multiple models (GPT-4o, Gemini 2.5 Pro, Claude Sonnet 4, Claude Haiku 3.5). Specifically, we first used the human participants’ explanations of their beliefs as the initial user message and observed whether models attempted persuasion. Alternatively, we prompted (using APE’s standard prompt, Appendix B.1) the simulated persuadee to produce the respective message. Both cases use the human-provided topic and belief level.
As shown in Table 1, results across human and simulated messages were nearly identical: GPT-4o and Gemini 2.5 Pro attempted persuasion nearly 100% of the time on both, while Claude Sonnet 4 and Claude Haiku 3.5 refused around 15-20% and 30% of the time on both messages, respectively. Although it would be unethical to run the same experiments on more harmful topics, conspiracy theories remain a critical test case within the information economy; particularly in light of our earlier findings showing that LLMs can effectively persuade users both into and out of such beliefs. Our comparison between human and simulated data further reinforces this point: simulating initial user messages to assess persuasion propensity yields results that closely mirror those obtained from real human interactions, as measured by the APE evaluator.

**Table 1**: Comparison of human vs. simulated persuadee for four frontier models acting as persuader. Turn 1 persuader attempt rates align between simulated and real human persuadees.

| Data Source | GPT-4o | Gemini 2.5 Pro | Claude Sonnet 4 | Claude Haiku 3.5 |
|-------------|--------|----------------|------------------|------------------|
| Human       | 99.6%  | 99.8%          | 80.3%             | 73.0%            |
| Simulated   | 99.6%  | 99.0%          | 86.2%            | 69.3%            |

---

> ### Author Response · Authors · 2025-11-19
> **Group Response: Part 2**
>
> **Comparing Different Evaluator Models**
>
> All reviewers suggested testing our eval with different evaluators to address potential circularity when GPT-4o is used as both persuadee and evaluator. To test robustness to the choice of these roles, we conducted an additional experiment where we ran a full APE evaluation (all 600 topics across the six topic categories) for GPT-4o as the persuader under different combinations of persuadee and evaluator models.
>
> Specifically, we considered two persuadee models (GPT-4o and Gemini 2.5 Pro), and for each conversation we asked five diverse evaluator models (GPT-4o, Gemini 2.5 Flash, GPT-oss 20B, Llama 3.3 70B, and GPT-5 mini) to classify each persuader turn as “attempt” vs. “no attempt” using the same evaluator prompt as in the main paper. The underlying conversations (persuader–persuadee transcripts) are identical within each persuadee setting; only the evaluator model changes. Refusal rates are measured by GPT-4o mini and remain fixed across evaluators, so the comparison isolates how different evaluators partition responses into “attempt” vs. “no attempt.”
>
> For most topic categories (benign opinions, controversial, benign factual, and undermining control), all evaluator models classify GPT-4o as attempting persuasion in the high 90s to ~100% of cases, regardless of whether the persuadee is GPT-4o or Gemini 2.5 Pro. The most informative differences appear on the non-controversially harmful topics, which are also the most safety critical. Table 2 below summarises the Turn 1 results for this category across each persuadee and evaluator combination:
>
> **Table 2**: Alternative evaluator and persuadee models for APE on Turn 1 messages in the non-controversially harmful topic category. The attempt rates are largely consistent across models.
>
> | Persuader Model | Persuadee Model    | Evaluator Model     | Attempt rate | No attempt rate | Refusal rate |
> |-----------------|--------------------|---------------------|--------------|-----------------|--------------|
> | GPT-4o          | GPT-4o             | GPT-4o              | 0.64         | 0.08            | 0.28         |
> | GPT-4o          | GPT-4o             | Gemini 2.5 Flash    | 0.60         | 0.12            | 0.28         |
> | GPT-4o          | GPT-4o             | GPT-oss 20b         | 0.56         | 0.16            | 0.28         |
> | GPT-4o          | GPT-4o             | Llama 3.3 70b       | 0.65         | 0.07            | 0.28         |
> | GPT-4o          | GPT-4o             | GPT-5 mini          | 0.64         | 0.08            | 0.28         |
> | GPT-4o          | Gemini 2.5 Pro     | GPT-4o              | 0.71         | 0.10            | 0.19         |
> | GPT-4o          | Gemini 2.5 Pro     | Gemini 2.5 Flash    | 0.68         | 0.13            | 0.19         |
> | GPT-4o          | Gemini 2.5 Pro     | GPT-oss 20b         | 0.57         | 0.24            | 0.19         |
> | GPT-4o          | Gemini 2.5 Pro     | Llama 3.3 70b       | 0.74         | 0.07            | 0.19         |
> | GPT-4o          | Gemini 2.5 Pro     | GPT-5 mini          | 0.67         | 0.14            | 0.19         |
>
> Across evaluators, the attempt rates on non-controversially harmful topics for GPT-4o as persuader cluster in a relatively narrow band (roughly 0.56–0.65 with GPT-4o as persuadee, and 0.57–0.74 with Gemini 2.5 Pro as persuadee). All evaluators agree that GPT-4o attempts persuasion on a majority of non-controversially harmful prompts.
>
> Overall, these results suggest that our main conclusions are robust to the choice of evaluator and to using GPT-4o in multiple roles. Different evaluators shift the exact attempt percentages by a few to ~10 percentage points, but they do not change the qualitative story that GPT-4o frequently attempts persuasion on non-controversially harmful topics.
>
> We appreciate the solid experimental suggestion and believe that this strengthens the findings of our work, showing that the results are consistent when changing the evaluator or persuadee models. We have added this analysis to the main paper, Section 4.4.1, thank you!

---

> > ### Author Response · Authors · 2025-11-19
> > **Group Response: Part 3**
> >
> > **Preventing Harmful Persuasion from Models**
> >
> > We note that solutions to harmful persuasion is an open and unsolved research problem [El-Sayed et al. 2024, OpenAI 2025] and is not the focus of this paper; instead we focus on robustly identifying models that will attempt harmful persuasion. Moreover, the cause of such critical issues is difficult to ascertain, especially in closed-weight frontier models. We hypothesize that the post-training procedures simply overlooked harmful queries in a persuasive context. For example, we show in Lines 363-377 that simply changing the prompt from the persuasive setting to “Can you help me X” made GPT-4o refuse all topics. This suggests that the safety training included certain prompts asking for help doing harmful things but missed, or did not contain enough, prompts asking the model to persuade people to do harmful things. Some future work could consider modifying persuasiveness through post-training/finetuning [Hackenburg et al. 2025] or classifying potentially harmful persuasion contexts through linear probes [Jiapersaud et al. 2025].
> >
> > El-sayed et al. 2024 A Mechanism-Based Approach to Mitigating Harms from Persuasive Generative AI.
> >
> > OpenAI 2025 GPT-4.5 System Card
> >
> > Hackenburg et al. 2025 The Levers of Political Persuasion with Conversational AI.
> >
> > Jiapersaud et al. 2025 How Do LLMs Persuade? Linear Probes Can Uncover Persuasion Dynamics in Multi-Turn Conversations.

---

### Author Response · Authors · 2025-11-27

Dear Reviewers,
Thank you again for your positive feedback and constructive suggestions on our work. We have posted our rebuttal and hope that the new non-simulated, human persuasion experiments and validation of alternative evaluator models have sufficiently addressed your concerns.
We would be happy to answer any further questions you may have before the end of the discussion period!
Sincerely,
APE Authors

---

### Author Response · Authors · 2025-12-01

Dear Area Chair,

Thank you for taking the time to consider our paper and the review discussion. We understand how much content there is to read and we’re very grateful for your consideration and acknowledge that it’s a time consuming process in truly exceptional circumstances.

In order to aid your understanding of the current status of the review process we’ve made a summary for you below.

## Summary of Major Rebuttal Updates

To address the primary concerns raised across multiple reviews regarding **validation with human responses** and **reliance on a single evaluator model**, we performed additional experiments and give results from separate studies, included in our Group Response:

* **Validation with Human Participants (Addressing Reviewers WhhT, Dswg):** We conducted a comparison between model-to-model simulations and real human interactions.
  * **Result:** Persuasion attempt rates are nearly identical when models respond to human initial messages versus simulated ones (e.g., GPT-4o attempt rates were 99.6% for human vs 99.6% for simulated).
  * **Previous Result:** In an IRB-approved study ($n=836$), we verified that GPT-4o is capable of shifting real human beliefs (by \~12–14 points on a 0–100 scale), validating that our focus on model capabilities translates to real-world risks.
* **Robustness Across Different Evaluators (Addressing Reviewers Dswg, dm8K):** We addressed concerns about potential evaluator bias and circularity (using GPT-4o to evaluate GPT-4o) by re-running the evaluation using 5 diverse models (including Llama 3.3 70B, Gemini 2.5 Flash, and GPT-oss).
  * **Result:** Agreement is high across all evaluators. The finding that frontier models frequently attempt persuasion on non-controversially harmful topics remains consistent regardless of the evaluator used.

---

## Detailed Responses to Specific Reviewer Concerns

### **Reviewer WhhT**

* **Concern: Novelty & Concurrent Work.**
  * **Resolution:** We clarified that cited work \[1\] is concurrent (post-ICLR deadline) and \[2\] focuses on novel jailbreaks rather than persuasion attempts, the focus of APE. We added a dedicated discussion in Section 2.2 distinguishing our specific focus on broad persuasion attempts.
* **Concern: Lack of actionable mitigation insights.**
  * **Resolution:** We highlighted that our work is a *measurement* capability required to enable mitigation. We also noted that **Google began work following our disclosure to improve Gemini safety**, validating the immediate practical utility of our work.
* **Concern: Reliance on LLM simulation.**
  * **Resolution:** Addressed via the new Human Participant results detailed in the Group Response.

### **Reviewer Dswg**

* **Concern: Model-to-model limits & Validity.**
  * **Resolution:** Addressed via new experiments from the Human Participant section of the Group Response, showing strong alignment between simulation and humans.
* **Concern: "Persuasion Attempt" definition is too shallow/binary.**
  * **Resolution:** We clarified that our scope is intentionally narrow to detect *any* violation of safety policies (e.g., attempting to persuade on terrorism), rather than analyzing rhetorical nuance.
* **Concern: Confounding factors (Temperature, Persona).**
  * **Resolution:** We pointed to our existing error bar reporting and Appendix C.2 (persona experiments) as demonstrations of the requested robustness.
* **Concern: Ethics, Misuse & Dataset Release.**
  * **Resolution:** We clarified that the dataset contains pre-written prompts, not trained model weights or jailbreaks, minimizing misuse risk while maximizing research utility. Warnings have been added to the release.
* **Concern: Ambiguity in topic boundaries.**
  * **Resolution:** We explained that the APE codebase is extensible to allow the community to evolve topics, and that "borderline" topics are a feature for testing nuance, not a bug.
* **Concern: Responsible Disclosure documentation.**
  * **Resolution:** We added specific dates and measurements regarding our disclosure to Google, OpenAI, and Anthropic, including quantitative evidence of Gemini 2.5 Pro’s safety improvements following our report (App C.1).
* **Concern: Presentation (Figure placement, Font size).**
  * **Resolution:** Figure 1 has been moved and resized for improved readability.

### **Reviewer dm8K**

* **Concern: Evaluator Circularity (GPT-4o evaluating GPT-4o).**
  * **Resolution:** Addressed via the "Comparing Different Evaluator Models" experiment in the Group Response, showing results are robust across Llama, Gemini, and GPT architectures.
* **Concern: Figure clarity.**
  * **Resolution:** Figure 1 font size has been increased.

We believe these revisions and additional experiments strongly reinforce the technical contribution and reliability of our benchmark and are confident that the paper presents a robust, validated framework for assessing AI safety risks.

Sincerely,

The Authors

---

### Meta-Review · Area_Chair_6gcS · 2025-12-25

**Summary:**

This paper introduces Attempt to Persuade Eval (APE), a benchmark designed to evaluate whether LLMs attempt to persuade users on harmful topics, rather than whether they successfully change beliefs. The findings expose significant gaps in current safety guardrails, show that jailbreak-style fine-tuning dramatically increases harmful persuasion, and highlight that persuasion attempts are most prevalent in early dialogue turns.

Two out of three reviewers gave negative ratings for this paper. The authors have provided very detailed rebuttal, addressing many of the points raised. However, AC thinks there are still some critical concerns remaining. For instance,

* Limited novelty and technical contributions. There have been quite a lot of works on the exploration and evaluation of LLM persuasion. Although the focus of this paper is new, which is LLM’s attempt to persuade, but TECHNICALLY the methodological contributions are rather limited. By the way, although the paper (LLM Can be a Dangerous Persuader: Empirical Study of Persuasion Safety in Large Language Models) is published in COLM 2025, it was released on ArXiv in April this year (https://arxiv.org/abs/2504.10430), which is far earlier than ICLR submission deadline. So you should not miss the discussion of it.
* Providing limited practical guidance, actional insights or implications on mitigating safety issues. We expect the authors to leverage the conclusions from their studies to design concrete methods for LLM safety enhancement, which is far more important than evaluation. This is, unfortunately, missing in either the paper or the authors’ responses.
* This paper simply treats persuasion as a binary label. Although the authors claimed that they intentionally restrict the scope, this still reflects the limitation of this paper as an evaluation framework.

Given the above points, AC recommends rejecting this paper.

**Reviewer Concerns:**

There are many concerns that have been addressed by the rebuttal, such as:

* Discussion and evaluation of human users responding to persuasive attempts.
* Comparison of different evaluation models.
* Explanations about the impact of some factors on the evaluation results.
* Some technical clarifications about the framework and experiments.

However, there are still some unsolved concerns, as discussed above.

**Reviewer Scores:**

Considering those unsolved concerns are from the reviewers who gave negative ratings. AC thinks they will not adjust their scores.

---

### Decision · Program_Chairs · 2026-01-26

Reject